# Computational Modeling of Pharmaceuticals with an Emphasis on Crossing the Blood–Brain Barrier

**DOI:** 10.3390/ph18020217

**Published:** 2025-02-06

**Authors:** Patrícia Alencar Alves, Luana Cristina Camargo, Gabriel Mendonça de Souza, Márcia Renata Mortari, Mauricio Homem-de-Mello

**Affiliations:** 1In Silico Toxicology Laboratory (inSiliTox), Department of Pharmacy, Health Sciences School, University of Brasilia, Brasilia 71910-900, Brazil; alencar.patriciaal@gmail.com (P.A.A.); gabrielms01@outlook.com (G.M.d.S.); 2Psychobiology Laboratory, Department of Basic Psychological Processes, Institute of Psychology University of Brasilia, Brasilia 71910-900, Brazil; luana.camargo@unb.br; 3Neuropharmacology Laboratory, Department of Physiological Sciences, Institute of Biological Sciences, University of Brasilia, Brasilia 71910-900, Brazil; mmortari@unb.br

**Keywords:** blood–brain barrier, in silico drug design, quantitative structure–activity relationship, machine learning, molecular docking, molecular dynamics, end-point free energy, prodrug design, CNS-targeted nanoparticles, neuroactive peptides, membrane transporters, central nervous system drugs

## Abstract

The discovery and development of new pharmaceutical drugs is a costly, time-consuming, and highly manual process, with significant challenges in ensuring drug bioavailability at target sites. Computational techniques are highly employed in drug design, particularly to predict the pharmacokinetic properties of molecules. One major kinetic challenge in central nervous system drug development is the permeation through the blood–brain barrier (BBB). Several different computational techniques are used to evaluate both BBB permeability and target delivery. Methods such as quantitative structure–activity relationships, machine learning models, molecular dynamics simulations, end-point free energy calculations, or transporter models have pros and cons for drug development, all contributing to a better understanding of a specific characteristic. Additionally, the design (assisted or not by computers) of prodrug and nanoparticle-based drug delivery systems can enhance BBB permeability by leveraging enzymatic activation and transporter-mediated uptake. Neuroactive peptide computational development is also a relevant field in drug design, since biopharmaceuticals are on the edge of drug discovery. By integrating these computational and formulation-based strategies, researchers can enhance the rational design of BBB-permeable drugs while minimizing off-target effects. This review is valuable for understanding BBB selectivity principles and the latest in silico and nanotechnological approaches for improving CNS drug delivery.

## 1. Introduction

The blood–brain barrier (BBB) is a highly selective semi-permeable membrane barrier located at the interface between the blood and central nervous system (CNS) tissue. It plays a relevant role in controlling the exchanges between these two compartments, allowing only specific molecules or ions to pass through either diffusion or through particular means such as facilitated diffusion, active transport, or passive transport. In this way, it is responsible for creating and maintaining the homeostasis of neuronal functions, coordinating communication between the periphery and the CNS, supplying nutrients to the brain, and protecting against toxic substances [1,2].

The BBB is an important limiting factor for drug delivery to the brain. Molecules’ ability to cross the BBB is limited by factors linked to their permeability, transport, and stability in the brain environment. Thus, this barrier is considered one of the main hindrances to permeation, protecting the CNS against toxic chemicals but slowing or inhibiting access to several therapeutic drugs [1,2].

For a molecule to reach the brain and exert its effects, it must possess specific characteristics that allow it to cross this selective barrier. One of the main characteristics is molecular size, as large molecules have difficulty crossing the BBB due to the tight junctions between endothelial cells (exceptions made for the bigger molecules that can be transported through transcytosis). Molecules with a molecular weight below 400–500 Daltons generally find it easier to pass through. Lipophilicity is another crucial characteristic, as the BBB comprises lipid bilayers of its constituent cells. Lipophilic molecules are more likely to cross the BBB since they can merge in cell membranes and cross them by passive diffusion.

Specific functional groups can also influence a molecule’s ability to cross the BBB. For example, hydroxyl and amine groups can increase water solubility, facilitating passage through endothelial cells. The molecule’s electrical charge also plays an important role, as charged molecules have more difficulty crossing the BBB due to the repulsion of charges in cell membranes. At the same time, specific transporters in the BBB can facilitate or hinder the passage of certain molecules. For example, P-glycoprotein (P-gp) is a vital transporter that can pump molecules out of endothelial cells, preventing their passage [1,2,3].

It is pretty challenging to evaluate all these parameters at the same time. Computational methods in drug discovery have made significant progress in the last decade. Researchers in the pharmaceutical industry and in universities have utilized calculation approaches for drug discovery and BBB crossing prediction [4]. This review aims to provide a comprehensive overview of the BBB and the computational techniques involved in developing CNS-targeted drugs.

## 2. BBB Morphology, Physiology, and Biochemistry

The BBB (Figure 1) is formed by the endothelial cells of the brain capillaries, which are specialized cells lining the blood vessels in the brain. These endothelial cells are unique due to the tight junctions between them, forming a physical barrier preventing many substances from passing from the blood into the brain [5]. Endothelial cell membranes are kept sealed by several proteins with transmembrane domains that prevent the leakage of liquids and solutes. The main protein groups related to this functionality are the occludins, claudins, and the junction adhesion molecules (JAMs) [6]. Additionally, the endothelial cells are supported by a basement membrane and are surrounded by pericytes, astrocytes, and glial cells that contribute to the integrity and function of the BBB [1,2].

Besides tight junctions, the BBB features selective active transport systems, allowing the controlled entry of essential nutrients such as glucose and amino acids through specific transporters. These mechanisms ensure that only substances necessary for proper brain function can access the CNS. A notable characteristic of the BBB is the absence of pinocytosis and phagocytosis in brain capillaries. This lack of capacity to absorb liquid and solid particles reduces the entry of undesirable substances, protecting the brain against toxic agents and pathogens [3]. On the other hand, brain endothelial cells are capable of adsorptive–mediated transcytosis. It works for cationic bigger molecules such as peptides and proteins such as albumin, immunoglobulins, transferrin, insulin, leptin, and many others [7].

Another type of transport that helps to keep the brain’s integrity is performed by efflux pumps present in the membranes of the endothelial cells (Figure 2). These proteins are responsible for the selective transport of specific substances from the blood to the brain or vice versa. These transporters allow the BBB to maintain strict control over the chemical composition of the brain environment. Table 1 summarizes the primary transporters present in the BBB, their function, and crystallized structures access codes.

Besides all xenobiotic/toxin transporters, the BBB is also populated with a variety of other SLC transporters that can provide the passage of nutrients across the membranes (even if they can eventually be used to transport drugs), such as the following:-Glucose transporter GLUT-1 (SLC2A1) [8];-L-type amino acids transport (LAT) system (SLC7 and SLC43) [9];-Na^+^ dependent multivitamin transporter SMVT (SLC5A6) [10];-Equilibrative (ENT) or concentrative (CNT) nucleoside transporters (SLC29 or SLC28, respectively) [11];-Neurotransmitter transporters as GLUT2 (SLC6A13) [12];-Ionic transporters as NKCC1 (SLC12A2) [13,14].

The fine regulation exerted by these transporters is essential for cerebral homeostasis. When imbalances or pathologies affect the BBB, complications such as toxin accumulation in the brain or nutritional deficiencies can occur [3,15].

The selectivity that protects the CNS also poses a challenge for treating neurological diseases, as many drugs cannot cross the BBB in therapeutic quantities. In this context, strategies to bypass the BBB, such as developing prodrug pharmaceuticals or using direct administration techniques into the CNS, have been the subject of intense research [1,2].

**Table 1 pharmaceuticals-18-00217-t001:** Main xenobiotic transporters present in the BBB.

Pump	Family	BBB Function	PDB ID ^1^	Ref.
MRP ^2^1	ATP-binding cassette (ABC)	Mediates ATP-dependent transport of glutathione and glutathione conjugates, leukotriene C4, estradiol-17-beta-o-glucuronide, methotrexate, antiviral drugs, and other xenobiotics.	2CBZ 4C3Z	[15,16,17]
MRP2		Transports a wide variety of conjugated organic anions, such as sulfate-, glucuronide,- and glutathione (GSH)-conjugates of endo- and xenobiotic substrates.	8JX7	[15,17]
MRP4		Mediates ATP-dependent transport of glutathione and glutathione conjugates, nucleosides, and analogs (e.g., antiviral drugs).	8BJF 8I4A8IZ7	[15,16,17,18]
MRP5		It acts as a general glutamate conjugate and analog transporter. It can limit the brain’s levels of endogenous metabolites, drugs, and toxins. Able to transport several anticancer molecules, including methotrexate and nucleotide analogs in vitro.	NA	[15,17,18,19]
P-gp ^3^		The best-known BBB transporter entity. Translocates drugs and phospholipids across the membrane. ATP-dependent efflux pump responsible for decreased drug accumulation in the brain and multidrug-resistant cells. More than 500 pharmaceuticals are already known as substrates [12].	6C0V 6FN1 6QEX 7A65 7O9W	[8,15,16,17,20,21]
BCRP ^4^		Actively removes a range of endogenous and exogenous substrates. It is mainly expressed at the luminal membrane of the BBB endothelial cells and displays substrate overlap with P-gp.	5NJ3 6ETI 6VXF 7NEQ 7OJ8 8BHT 8P7W	[17,22]
OAT ^5^1	Solute Carrier (SLC)	Involved in the transport of neuroactive tryptophan metabolites kynurenate (KYNA) and xanthurenate (XA) and may contribute to their secretion from the brain.	8BVR (rat, kidney)	[23,24]
OAT3		It plays a role in the efflux of drugs and xenobiotics, helping reduce their undesired toxicological effects on the body. Mediates the transport of p-aminohippurate, benzylpenicillin, and the statins pravastatin and pitavastatin from the cerebrum into the blood circulation across the BBB.	NA ^8^	[15,23,24]
OCTN ^6^2		Transport organic cations and carnitine.	NA	[25,26]
OATP ^7^1A2		It plays roles in the blood–brain and cerebrospinal fluid barrier transport of organic anions and signal mediators and in hormone uptake by neural cells.	NA	[27]
OATP1C1		Facilitates the transport of thyroid hormones across the blood–brain barrier into glia and neuronal cells in the brain.	NA	[15,17]
OATP2B1		Mediates the uptake of the neurosteroids DHEA-S and pregnenolone sulfate into the endothelial cells of the BBB as the first step to enter the brain.	NA	[15,17,27]
OATP3A1		Recognizes various substrates, including thyroid hormone L-thyroxine, prostanoids such as prostaglandin E1 and E2, bile acids such as taurocholate, glycolate, and glycochenodeoxycholate, and peptide hormones such as L-arginine vasopressin, likely operating in a tissue-specific manner.	NA	[17,28]

^1^ PDB ID: unique four-character alphanumeric accession code provided by the Protein Data Bank; ^2^ MRP: multidrug resistance-associated protein; ^3^ P-gp: P-glycoprotein 1; ^4^ BCRP: breast cancer resistance protein; ^5^ OAT: organic anion transporter; ^6^ OCTN: organic cation/carnitine transporter; ^7^ OATP: organic anion-transporting polypeptide; ^8^ NA: not available.

Membrane transporters are relevant in the function of the blood–brain barrier. They are an important obstacle to overcome by the molecules that have to enter the cerebral tissue. This selectivity is needed to protect the CNS from potentially harmful substances and maintain an environment conducive to healthy brain function. At the same time, they are interesting targets to be evaluated when planning drugs to act in the CNS.

## 3. Computational Prediction Models for BBB Crossing

The prediction of molecule passage through the BBB is a big issue in biomedical research, especially when designing drugs for treating neurological diseases. Various prediction models have been developed to study the behavior of substances when interacting with the BBB. These models include computational methods, in vitro and in vivo experiments, all of them contributing to a better understanding of the factors influencing cerebral permeability [4].

### 3.1. Computer-Aided Drug Design Methods

Drug design is an approach to the discovery and development of new drugs. Two interesting approaches in this context are ligand-based drug design (LBDD) and structure-based drug design (SBDD), both categorized as methods of computer-aided drug design (CADD) [29].

In LBDD, drug design is guided by molecules that bind to a desired target protein. This method is based on the principle that similar chemical scaffolds may have similar biological activity. It is useful when the structure of the target is not available [30]. The most common and fundamental steps of LBDD are usually headed by data collection [31,32], since the information about bioactive ligands and their biological activity must be gathered and analyzed beforehand. These ligands can be known chemical compounds or derived from screening experiments.

After the compound library is set, the activity (or property) quantification can begin. Quantitative relationships between the chemical structure of ligands and their biological activity or property are established. Statistical and Machine Learning methods are often employed at this stage. The results obtained will set the foundations for the model building. Mathematical or statistical models, such as quantitative structure–activity/property relationship (QSAR/QSPR) models, are developed to predict the biological activity of new compounds based on their structural characteristics. New compounds can be designed based on the optimized model to improve the desired activity [31,32].

The focus of the SBDD methods is on the three-dimensional structure of the target (mainly proteins). Detailed information about the protein’s geometry and binding sites is needed to guide the design of the ligands to be tested. Three-dimensional protein structures are used to make binding predictions using Molecular Docking (MDo), providing information about the chemical nature of the binding process. A kinetic perspective of the molecular interactions over time using the three-dimensional structures of the protein and the proposed ligand is achievable using molecular dynamics (MD). These tools facilitate understanding the target structure and guide the design of medicinal molecules. The SBDD approach requires, in the first place, the determination of the protein structure [30,31].

Techniques such as X-ray crystallography or nuclear magnetic resonance are employed to obtain the three-dimensional structure of the target protein. The coordinates and experiment details can be deposited on the Worldwide Protein Data Bank https://www.wwpdb.org/ (accessed on 25 November 2024), which manages the repository of information about the 3D structures of proteins, nucleic acids, and complex assemblies [33]. These structures can be searched and downloaded through one of its members, such as the Research Collaboratory for Structural Bioinformatics Protein Data Bank—RCSB PDB—https://www.rcsb.org/ (accessed on 25 November 2024) [34].

The next step in SBDD aims to determine the binding sites on the protein where ligands interact. These sites are often pockets or cavities on the protein’s surface. It is common to find crystallographic structures obtained with an agonist or inhibitor bound to the active or allosteric sites [30].

After the identification of the binding site, the most common technique used in SBDD is Molecular Docking (MDo). Candidate molecules are modeled and virtually “docked” into the active sites of the protein, simulating molecular interaction. This can provide an initial prediction of binding affinity and geometry. There are issues related to the induced-fit result obtained by conventional docking methodologies, and simple scoring functions cannot ideally deal with conformational changes that may occur in proteins upon ligand binding [35]. When it is complicated to predict the interaction between the target and ligands, or when there is an elevated inaccuracy, an enhanced sampling approach or all-atoms force fields (like Molecular Mechanics Poisson–Boltzmann (or generalized Born) surface area (MM/PB(GB)SA)) can be applied to obtain better results [35,36,37,38].

Based on the docking results, molecules can be chemically optimized to improve interaction with the target protein.

Both LBDD and SBDD are complementary and often used together (Figure 3) to maximize the efficiency of drug design. CADD, in general, has significantly accelerated the drug discovery process, making it more effective and economical [39].

#### 3.1.1. Quantitative Structure–Activity/Property Relationship

QSAR or QSPR models use statistical algorithms to correlate the physicochemical and structural characteristics of a molecule with its pharmacological characteristics. Using this approach, researchers can predict how modifications in the molecular structure of a compound will affect its properties and biological behavior. QSAR/QSPR methods are based on the premise that the molecular structure of a substance is inherently linked to its properties and, consequently, to its biological effects. Based on this relationship, it is possible to develop mathematical models to quantify this correlation. These models are decisive for the rational optimization of compounds, saving time and resources by directing experimental synthesis toward substances with a higher likelihood of success [40,41,42].

QSAR/QSPR models can be classified according to their dimensional perspectives: one-dimensional (1D), two-dimensional (2D), and three-dimensional (3D). The first dimension covers the simplest molecule characteristics, such as molecular weight or the number of carbon atoms. In 2D models, structural information is represented by considering features such as functional groups, chemical bonds, topological indices, electrostatic parameters, substructural fragments, and other molecular descriptors. In 3D models, the molecule’s three-dimensional geometry is considered, encompassing descriptors such as molecular volume, surface, HOMO, and LUMO (highest-occupied and lowest-unoccupied molecular orbitals), providing a more realistic representation of molecular interactions at the atomic level [43]. More complex and variable approaches based on molecular dynamics and ligand–protein interactions have already been proposed, such as 4D [44], 5D [45], or even 6D [46]. However, the increase in complexity leads to an exponential computational cost. Thus, most QSAR models published in the literature are based only on 1D and 2D descriptors, and a few models use both 2D and 3D molecular descriptors [44]. In the real world, bioactive compounds may have complex geometries and be highly flexible due to rotatable bonds, which cannot be measured using 1D or 2D descriptors. It is possible, however, to include 3D parameters through one or more unique ligand conformations, combining this information with its activity.

Model validation is a needed and important process to test the predictive ability of the algorithm, using independent data [40,41,42].

The Organization for Economic Co-operation and Development (OECD) produced a document to provide guidelines for validating QSAR models. It was proposed to help strengthen the reliability and acceptance of QSAR models by regulatory agencies worldwide. Even though it was developed to serve the development of pesticides and chemical substances in general, its precepts are coherent and applicable to the QSAR in general, including the development of pharmaceuticals. The guideline is divided into five general principles, which would make a QSAR method reliable and valid for general purposes [47].

The OECD principles for the development of a QSAR model are:A defined endpoint;An unambiguous algorithm;A defined domain of applicability;Appropriate measures of goodness-of-fit, robustness, and predictivity;A mechanistic interpretation, if possible.

The endpoint “BBB permeability” can be either categorical (BBB+ or BBB-) or numerical (logBB values—Equation (1)). Most of the data regarding this endpoint are categorical, which means that the information are divided into only two groups, one containing molecules that can permeate through the membrane and the other comprising molecules that cannot. Closer to reality, the parameter logBB informs the logarithm of the brain–plasma concentration ratio, which allows the quantitative evaluation of the permeability [48].(1)$$\log\mathrm{BB}=\frac{\left[drug\;in\;the\;brain\right]}{\left[drug\;in\;blood\right]}$$

Another numerical data available for the evaluation of the BBB permeation are the logPSs (permeability surface-area products), which are considered more informative measures, but they demand lots of resources and are more difficult to obtain. It is often based on an in vivo perfusion methodology. Thus, the logPS measurement is resource-intensive and relatively low throughput [49]. Consequently, there is a notable lack of literature information available on this topic, making this variable considerably less used. The PS can be determined using the Renkin–Crone [50,51] Equation (2).(2)$$\mathrm{PS}=-F\;\ln\left(1-\frac{K_{\mathrm{in}}}{F}\right)$$
where PS is the permeability surface-area product (in mL/min/g brain), F is the perfusion flow rate (mL/min/g brain), and K_in_ is the transfer constant, obtained from the in vivo experiments after the measurement of the compound concentrations in the brain and in the fluid.

The development of a QSAR/QSPR model requires, at first, the collection of experimental data, including information on compounds’ molecular structure, physicochemical characteristics, and biological activities. Next, relevant molecular descriptors must be selected, serving as independent variables in the models. Statistical and mathematical techniques are used to develop algorithm models that best represent the relationship between molecular descriptors and biological activity.

The mathematical quantification of the structure–activity relationship or the structure–property relationship is performed using techniques such as partial least square regression (PLS), multiple linear regression analysis (MLR), principal component regression analysis (PCR), K-nearest neighbors, or canonical correlation analysis [52], among others. Usually, PLS is applied to predict one set of dependent variables using a batch of independent variables. This approach can correlate the interdependencies between two series of multiple related variables. MLR is a usual mathematical pathway to predict the relationship between molecular structure and toxicity [53]. PCR is often used to extract the principal components of independent variable groups and evaluate the regression [54].

The scope and limitations of a model must be clearly defined based on the structural, physicochemical, and response information in the model training set. A model can only reliably predict chemicals similar to those used in its development. Predictions outside this scope are improbable to be reliable. When using a QSAR/QSPR, it is imperative to assess whether it falls within its applicability domain and determine the known boundary. This assessment can be made categorically or quantitatively, with a confidence interval to precisely determine the similarity between the chemical of interest and the model training set [47].

Some BBB permeability models have already been developed and made freely or commercially available online or as a part of chemistry software. Each one has its own mechanism to verify the domain boundaries of their analysis. SwissADME http://www.swissadme.ch/ (accessed on 25 November 2024) uses a statistical model based on Monte-Carlo optimization of what they call a “boiled egg” (Brain Or IntestinaL EstimateD permeation method; Figure 4a), which summarizes the results of the analysis in an elliptical graph that resembles an egg. It results from evaluating a training set containing 260 molecules (156 BBB permeant and 104 non-BBB permeant) with reliable measurements of blood–brain partition (log BB). For the comparison, they used two descriptors, the octanol/water partitioning coefficient (logP) and the topological polar surface area (TPSA), both 2D-QSAR descriptors [55]. This tool was integrated into another analysis to inform if the query molecule can be a P-gp substrate, a prediction that can directly impact the possibility of the drug permeating the BBB. They use a machine learning (ML) method to obtain this information, with a training set including 1033 molecules. The result is a categorical “yes” or “no”, helping to increase the reliability of the prediction [56].

Several other models have already been proposed. Most are based on 2D-QSAR properties. Chemaxon’s Calculator is equipped with two different tools (Figure 4b). CNS multiparameter optimization (MPO) score is based on a training set of 108 CNS marketed drugs, using the following physicochemical parameters: i—lipophilicity, calculated partition coefficient (ClogP); ii—calculated distribution coefficient at pH 7.4 (ClogD); iii—molecular weight; iv—TPSA; v—number of hydrogen-bond donors (HBDs); and vi—most basic center (pKa). Analyzing these parameters involves evaluating the parameter value “desirability”, i.e., the best range of values typically present in SNC drugs [57,58,59]. The other Chemaxon’s Calculator is the ’BBB score’, based on a QSAR/QSPR algorithm that uses stepwise and polynomial piecewise functions, calculated after five physicochemical descriptors: i—number of aromatic rings; ii—heavy atoms; iii—MWHBN (a descriptor comprising molecular weight, hydrogen bond donor, and hydrogen bond acceptors); iv—TPSA; and v—pKa [60]. Both methods have a similar score system, considering the molecule as BBB permeable if the result is ≥4.

More recently, a more significant dataset of numeric (n = 1508) and categorical (n = 7505) experimental BBB permeation results has been developed [48]. The sample size is relevant to the model’s accuracy. Indeed, commercial software already includes a QSAR/QSPR method based on these results [61], reporting an accuracy of over 0.8 and a sensitivity of over 0.9 for their training set.

QSAR/QSPR models for predicting BBB passage are commonly applied in the initial screening of compound libraries, ranking those more likely to cross the barrier (or those that the BBB will most probably withhold if the intention is to develop some drug that may be neurotoxic). These models are also used to help the structural optimization of lead compounds, guiding chemical modifications that can change brain permeability. However, there are significant challenges. The complexity of the BBB and interindividual variability in its permeability can limit the accuracy of the models. Additionally, the need for high-quality experimental data and the proper choice of descriptors are critical factors that impact the reliability of QSAR/QSPR models [62].

Pedagogically, there are three main steps needed to develop QSAR/QSPR models to predict the BBB permeation:

*Data Collection and Selection of Training Sets:* Usually, the first step when developing a QSAR/QSPR model is collecting experimental data on the passage of compounds through the BBB. These data can be obtained from in vitro or in vivo studies. There are also pharmacokinetic databases that can be useful in this task. The diversity and representativity of the dataset are relevant to increasing the model’s robustness.

*Selection of Molecular Descriptors*: Molecular descriptors may include physicochemical properties (such as logP, solubility, and molecular weight), topological characteristics (such as connectivity indices), electronic parameters (such as dipole moments), or pharmacodynamic characteristics (such as P-gp affinity). The correct choice of descriptors is directly related to the accuracy of the prediction and can be guided by prior knowledge of medicinal chemistry, BBB biology, or statistical selection.

*Model Construction and Validation*: Statistical techniques such as multiple linear regression, discriminant analysis, and machine learning methods like neural networks, support vector machines, and random forests are commonly used to build QSAR/QSPR models. Model validation involves dividing the dataset into training and test subsets, using techniques such as cross-validation to assess the model’s predictive ability.

QSAR/QSPR models have broad applications in the pharmaceutical industry, predictive toxicology, and other chemical and biological research areas. In an increasingly data-driven research landscape, QSAR/QSPR will continue to be a useful tool for the advancement of medicinal chemistry [40].

#### 3.1.2. Molecular Docking (MDo)

MDo is a computational technique that simulates molecular interactions between a drug and some target (like a BBB macromolecule), helping to predict affinity and providing tools to understand the likelihood of passage. This technique is often used in drug discovery as a tool to model the interaction between a ligand (usually a chemical compound, mostly small molecules) and a receptor (frequently a protein). The MDo process involves a series of intricate and interconnected steps. Initially, both the ligand and the receptor undergo preparations, such as geometry optimization and charge assignment, to accurately represent their three-dimensional structures. Next, a search space is defined that delimits the possible orientations and positions of the ligand relative to the receptor’s binding site. MDo is applied in various areas of biomedical research, including drug discovery and drug design. Its ability to virtually screen large compound libraries can save time and resources [40,41,42,63].

The effectiveness of MDo largely depends on the search algorithms used, which explore this conformational space in search of the most energetically favorable configuration. During this process, various factors are considered, including van der Waals interactions, electrostatic forces, hydrogen bonding, and other components of the binding energy. Sophisticated algorithms (e.g., genetic algorithms, Monte-Carlo simulated annealing, shape-matching algorithms, or incremental construction) seek to minimize the interaction energy, providing a more realistic representation of molecular interactions [64].

The generated conformations are evaluated through scoring functions, which assign values to different ligand poses relative to the receptor. These functions consider the stability of the interaction and are used to select the most promising conformations for subsequent analyses. The accuracy of MDo depends on the quality of the receptor’s three-dimensional structure and the predicted pose. Because MDo is a technique that involves a certain protein rigidity, it is common to use improvement tools to obtain better results. Molecular dynamics (MD) simulations, a computationally more expensive technique, can be used with MDo during protein preparation to generate a protein after relaxation and energy minimization for docking. MD also helps to refine the docked receptor–ligand structure and can also include the effect of solvents in the final simulation. In addition, it can generate free binding energy calculations that will serve as a reference for ranking the tested ligands [37].

Enhanced sampling of the protein pocket is another MD technique that improves the MDo results. It introduces changes in the protein that happen after ligand interaction and may alter the binding site’s conformation, improving the results’ accuracy [38].

MDo identifies specific molecular interactions that facilitate the interaction with some BBB proteins, which helps to understand the chemical modifications that can be made to lead compounds to enhance or inhibit brain permeability. The evaluation of the influence of BBB proteins in drug permeation is mainly performed using P-gp, which is already very well characterized and studied, with several PDB IDs deposited [63].

Despite its potential, MDo faces significant challenges in predicting BBB permeation. The accuracy of predictions is limited by the quality of available receptor structures and the complexity of molecular interactions involved in BBB passage. Additionally, docking models often do not consider the dynamics of the BBB in vivo, where factors such as blood flow, interaction with other cells, membrane viscosity, and several other parameters can influence permeability [49].

MDo can be combined with other computational and experimental approaches to improve prediction accuracy. QSAR/QSPR models, MD simulations, and experimental permeability data can be integrated to provide a more comprehensive view of molecular interactions at the BBB [65,66].

MD can help to understand better the interaction between ligands and the targets, such as P-gp, or even predict the diffusion of the small molecules through the membranes. However, this calculation is more computer-demanding, and its use is more common in academic research, where several methods have already been developed. The increasing computational development and spreading of GPU calculations are increasing the commercial use of MD techniques [49,67,68,69].

#### 3.1.3. Molecular Dynamics-Based Techniques

MD simulations can explain better the molecular interactions and dynamics involved in drug transport across the BBB, which can be used to predict the permeability of molecules based on their structural properties and interactions with the barrier components [49,70].

Passive diffusion is the main mechanism of BBB permeation for small and lipophilic molecules. MD simulations can use models of lipid bilayers, which are helpful to study the dynamic behavior of the interactions between the membrane and the drug. This type of simulation provides a deep understanding of how specific molecular characteristics influence BBB permeability [49,71]. The simulation results have shown a good correlation with in vitro results of permeability assays [49,68].

Active transport mechanisms at the BBB can also be investigated using MD simulations. The interaction of drugs with efflux transporters, such as P-gp, can be simulated at the molecular level, evaluating the influence of this process on the drugs’ CNS availability [72].

Drug delivery systems such as nanoparticles (NPs) and liposomes can also be MD simulated. These systems can be rationalized to improve drug transport across the BBB by enhancing the solubility and stability of drug molecules [73,74]. NPs with specific ligands (e.g., resveratrol) can increase their uptake by brain endothelial cells, improving drug delivery to the CNS [75].

MD simulations can be applied to larger molecules (e.g., biomolecules or nanocarriers) as well, but at a higher computational cost. Nine BBB-penetrating peptides have been studied using MD to characterize their physicochemical and dynamic behavior in physiological media. These peptides are known for their ability to cross the BBB and can be used as drug carriers [76]. Metallic NPs have also been studied in BBB membrane models, evaluating their kinetic permeability through the system by MD [67].

Molecular dynamics methods provide, thus, an interesting approach for evaluating drug permeation across the BBB for case studies. The simulation of the molecular interactions and dynamics involved in drug transport allows the prediction of the permeability of drug candidates and carrier systems with a deeper mechanistic understanding, providing useful and direct information regarding structural modifications that can bring the desirable permeability profile.

##### End-Point Free Energy

In recent years, end-point free energy techniques have become more prevalent in molecular modeling, particularly in drug design and simulations of biomolecular complexes. These techniques enable calculating the binding free energy between proteins and ligands without explicitly describing all the intermediate steps of the binding and dissociation processes. End-point methods are more superficial and computationally cheaper than more rigorous approaches like alchemical and pathway sampling methods. However, this simplicity often comes at the expense of accuracy, particularly in systems involving significant conformational movements or structural rearrangements [77,78,79,80].

End-point techniques consider two primary states: the bound state (ligand complexed to the protein) and the unbound state (protein and ligand separated in solution). The calculation assumes that the protein and ligand conformations in both states are sufficiently similar, allowing the exclusion of explicit intermediate states in the free energy calculation [78,80].

MM/PBSA (Molecular Mechanics Poisson–Boltzmann Surface Area) and MM/GBSA (Molecular Mechanics Generalized Born Surface Area) are the two most commonly used end-point methods. MM/PBSA uses Poisson–Boltzmann equations to calculate the electrostatic potential around the protein and ligand. It is particularly suited for systems where solvent effects, especially from water, are critical. MM/GBSA uses the Generalized Born approximation to calculate the electrostatic solvation contributions. It is less accurate than MM/PBSA but has the advantage of being computationally faster [78,80].

The binding free energy can be calculated using Equation (3).∆G = ∆H − T∆S = ∆E_MM_ + ∆G_sol_ − T∆S(3)
where ΔG is the Gibbs free energy variation, ΔH is the enthalpy variation, T is the absolute temperature in Kelvin, and ΔS is the entropy variation. The enthalpy variation can be decomposed in terms of molecular mechanics (MM) changes in energy (ΔE_MM_) and solvation free energy (ΔG_sol_). The MM energy is the sum of the variation in the internal (dihedral, angle, and bond energies), electrostatic, and van der Waals energies, respectively (ΔE_int_, ΔE_ele_, and ΔE_VdW_, respectively) (Equation (4)).∆E_MM_ = ∆E_int_ + ∆E_ele_ + ∆E_VdW_(4)

The ΔG_sol_ parameter represents the sum of the polar, electrostatic solvation energy (ΔG_PB/GB_), calculated according to the desired model. The nonpolar contribution to this parameter is represented by the term ΔG_SA_, obtained typically using the solvent-accessible surface area (SASA) approach [81], as shown in Equations (5) and (6).∆G_sol_ = ∆G_PB/GB_ + ∆G_SA_(5)∆G_SA_ = γSASA + b(6)

Usually, MM/PB(GB)SA techniques are performed within MD simulations of the protein−ligand complex using an explicit solvent model. After all MD snapshots are obtained, the solvent and ions are removed, and the solvation energy can be calculated after applying the PB(GS)SA solvent model. The solute’s entropic term of this equation can be obtained from a set of MD snapshots. The sum of the obtained energy components is the final binding free energy (Figure 5).

Some end-point free energy approaches have already been used to evaluate the permeation of compounds through membranes. The potential of mean force (i.e., the function of the free energy fluctuation along a determined surface in reference to a determined coordinate—an atom, a bond, or even a specific distance among atoms) has already been used to evaluate the permeation of 12 compounds with a known permeation profile, correlating well with both logBB (R^2^ = 0.94) and logPS (R^2^ = 0.90) [49].

Free energy surface calculations were used to predict compounds translocation rate and permeability through membranes during a simulation. Predictions using this technique can strongly or weakly correlate with in vitro or in vivo experiments, depending on the free energy surface of the tested compound and how it interacts with the membrane. The results suggest that the evaluation of the passive diffusion alone may be insufficient to reliably predict the BBB permeation of small molecules with different physicochemical properties. Other processes, such as the drug interactions with the cell membrane, sequestration within the cell, efflux, dissociation, or enzymatic degradation, events hard to evaluate experimentally, may significantly influence the permeation of the substances [82].

For compounds that can pass through the cell membrane more easily and are known negatives as P-gp substrates, the free energy calculations tend to be more reliable [68,78].

However, simulating biologically relevant systems that exhibit substantial conformational changes or alterations in solvation states is a heavy task. Advancements like the development of new entropy estimators and solvation correction techniques are contributing to increasing the precision of these simulations. The hardware improvement (provided by MD accelerated by GPU calculation, for example) is contributing to the spreading of these techniques as well. At the same time, these methods can yield rapid estimates of free energy, making them an interesting resource in extensive ligand screening processes, where finding a balance between accuracy and computational expense is important. Nevertheless, for systems that demand accuracy, more advanced techniques like full molecular dynamics simulations or alchemical methods should be considered [78,80].

#### 3.1.4. Machine Learning Models

Machine learning (ML) has been applied to predict BBB permeability. Models trained with large datasets containing information on chemical structure and substance permeability can make predictions based on patterns identified during training. Such models rely on the ability to analyze vast and varied datasets, exploring subtle patterns and complex relationships that escape conventional human perception [41,65].

ML algorithms, such as neural networks and decision trees, are used to train data ranging from genetic information to clinical test results. The amount of data can be enormous, allowing the algorithms to develop the ability to identify unusual patterns that would have been hard to find using conventional methods. This includes even the integration of genomic and proteomic data, which could help to identify molecular markers associated with brain permeability, shedding light on its underlying mechanisms. In this way, the union between data and machine learning transcends traditional methods, enabling a more precise and personalized approach to BBB prediction [41]. ML techniques have also empowered QSAR/QSPR models, increasing the assertiveness of the results [83].

As seen with the QSAR/QSPR approach, the initial ML prediction algorithms were based mostly on qualitative categorical variables (e.g., BBB+ or BBB-). Several techniques have already been used, mostly applying the ML algorithms to 2D and 3D chemical descriptors and commonly providing high accuracies, easily above 90% [84]. Usually, the research in this field uses more than one algorithm, so it is possible to compare the performance. The results, however, are dependent on the choice and quality of the dataset and descriptors chosen. Thus, random forest [85,86,87,88], XGBoost [89], support vector machines [87,90,91,92], decision trees [90,93], multilayer perceptron [86,87], linear discriminant analysis [94], artificial neural networks [90], k-nearest neighbors [87], genetic algorithms [91], or other approaches are cited as more or less accurate, depending on each data. The “consensus” approach uses combinations of training methods and classifiers, providing a more rational result, but as imagined, at a higher computational cost [86,87,92].

The most common continuous variable linked to BBB permeation is logBB (Equation (1)), which can also be used in ML techniques. Several different models using logBB and ML have already been constructed, including support vector machines, multiple linear regression [95], graph neural networks [96], artificial neural network [97,98], deep neural nets [99], and deep learning [100], among others, provide quantitative results.

Despite the advantages, the application of ML in BBB prediction faces several challenges. The quality and representativeness of the data are fundamental to the technique. The scarcity of high-quality experimental data can limit the effectiveness of the models. Additionally, the biological complexity of the BBB, involving dynamic and multifactorial interactions, can be challenging to capture fully in ML models [101].

#### 3.1.5. Challenges and Opportunities in CADD for BBB Permeability Prediction

The application of computer-aided drug design (CADD) in predicting blood–brain barrier (BBB) permeability has significantly advanced the field of CNS drug discovery. However, despite its potential, several challenges remain, limiting the accuracy and applicability of computational models. At the same time, new opportunities are emerging as computational power, algorithms, and experimental validation techniques evolve.

One of the primary challenges in CADD for BBB permeability prediction is the complexity and dynamic nature of the BBB. The barrier is influenced by various factors, including active transport mechanisms, efflux proteins, and disease-induced changes, which are difficult to capture using in silico models alone. Many existing prediction tools rely on simplified representations of molecular interactions, often neglecting time-dependent conformational changes and the role of physiological conditions. Additionally, the limited availability of high-quality experimental permeability data presents a challenge in training robust ML and QSAR models. Inconsistent or scarce logBB and logPS values in publicly available databases can lead to biased predictions, reducing the reliability of computational approaches [47].

Another major challenge is the gap between computational predictions and in vivo outcomes. Many computational models, particularly those based on molecular descriptors or MD, assume that passive diffusion is the primary transport mechanism. However, many CNS drugs rely on carrier-mediated uptake or face active efflux by P-gp and other transporters. As a result, models that do not account for these processes may yield misleading predictions. MD simulations and free energy calculations have improved the ability to study the passive diffusion mechanisms, enriching the understanding of this mechanism. Nevertheless, these techniques are still relatively computationally expensive, depending on the time lapse evaluated, and require careful parameterization [78,79,80].

Despite these challenges, significant opportunities exist for improving BBB permeability predictions using CADD. Artificial intelligence and deep learning advancements are opening new possibilities for data-driven modeling, allowing for better pattern recognition and multi-factorial analysis. The integration of large-scale BBB permeability datasets with AI-driven prediction models could enhance the accuracy and generalizability of computational approaches. Furthermore, hybrid modeling approaches that combine QSAR, molecular docking, MD simulations, and machine learning could provide a more holistic understanding of drug permeability mechanisms.

Another opportunity lies in the development of BBB-targeted drug delivery systems, including nanoparticles and prodrugs (better explained in the next topic). Computational tools can aid in the design of nanocarriers optimized for BBB transport, reducing experimental trial and error. Similarly, in silico predictions of enzymatic activation and transporter interactions can guide rational prodrug design, accelerating CNS drug development.

Combining molecular-level simulations with physiological pharmacokinetic and multi-scale modeling approaches offers a promising direction for bridging the gap between computational predictions and clinical outcomes. These models could enhance the translational impact of computational studies by incorporating physiological parameters such as cerebral blood flow, transporter expression levels, and disease-specific BBB alterations.

## 4. Combination Strategies for BBB Permeability Enhancement

### 4.1. Prodrug Design for Enhanced BBB Permeability

Prodrug design is a drug delivery strategy that uses a biologically inactive derivative of a compound that becomes pharmacologically active after undergoing enzymatic cleavage or chemical modification within the body. By modifying the lipophilicity or polarity of the drug, prodrugs can cross the BBB more efficiently by being carried by SLC transporters. This method has been experimentally applied to various drugs to improve their brain penetration and optimize pharmacokinetics. For example, the LAT system has already been studied as a transporter of prodrugs for ketoprofen, ibuprofen, nipecotic acid, and salicylic acid [102].

The prodrug approach has some advantages, such as easier synthesis, predictable in vivo results, and better safety profiles. However, the efficacy of prodrug strategies is dependent on the interaction between drug-metabolizing enzymes, efflux transporters, and the specificity of carrier systems at the BBB. Nevertheless, there is little comprehensive data on transporter structures and carrier systems specificity, further complicating the design of optimal prodrugs. The continued development of prodrug technologies holds significant promise for advancing CNS drug delivery and therapeutic strategies [103].

One of the most studied strategies is the modification of the drug structure using a glycoside group as a prodrug moiety. This approach aims to use the glucose transporter present in the BBB, increasing the drug permeation into the brain. Adjustments to the glycosidic bond orientation or the introduction of specific chemical groups can enhance the permeability and metabolic stability of the prodrugs [104,105].

### 4.2. Nanoparticle Delivery Systems

Due to their possible ability to penetrate the BBB, NP-based drug delivery systems are a particular way to distribute substances into the brain. Their small size allows them to cross the BBB, and they can be designed with specific surface modifications to enhance targeted delivery. NPs made from materials such as lipids, polymers, or metals can potentially deliver drugs that typically cannot cross the BBB, improving the bioavailability of therapies for brain-related conditions [106]. Several characteristics of the NPs can be modified to enhance delivery. Size, shape, and surface charge are among the most common properties evaluated during the development of an NP. These characteristics are important to control biodistribution and efficient targeting. As a result, improved therapeutic outcomes are expected [107,108,109].

NPs can cross the BBB through a variety of mechanisms. Receptor-mediated endocytosis, paracellular transport, and passive diffusion are described in the literature. The development of NPs coated with peptides, antibodies, or glycosides can further enhance their ability to deliver the drug across the BBB [110]. The production of NP-based CNS drugs has increased in the last few years, and several drugs are under clinical trials. Some of them have already been approved for diseases like schizophrenia (paliperidone palmitate nanocrystals), attention deficit hyperactivity disorder (methylphenidate), or cancer (paclitaxel nanomicelles). However, due to challenges inherently linked to NP production, most studies are still in phase 1 or 2 of clinical trials. Nevertheless, NP-mediated delivery holds great promise for enhancing brain drug delivery and targeting specific diseases, such as Alzheimer’s and brain cancers [111,112].

As already discussed in the MD topic, computational methods have already been used to evaluate NPs permeation through the BBB. Other properties, such as NP aggregation and NP cytotoxicity (membrane damage), have already been studied for some types of NPs. For example, magnetic NPs BBB permeation has already been studied through steered MD [67], allowing the obtaining of diffusion coefficients. The aggregation of functionalized carbon nanotubes, observed in in vitro studies, was successfully evaluated through MD, demonstrating the relevance of the Van der Waals interactions for the aggregate formation [113].

Dual-function drug delivery systems have emerged as a promising approach to improve BBB permeability while simultaneously encapsulating therapeutic agents for sustained release. These systems can enhance drug distribution within the brain and reduce side effects by enabling targeted delivery and minimizing peripheral exposure. Successful implementation of these systems has demonstrated improved therapeutic outcomes, such as enhanced anti-tumor therapy, by overcoming the permeability challenges of the BBB [13,114,115].

Incorporating chemical modifications and NP technologies, such as cubosomes and lipid–NP complexes, further enhances the ability to target CNS pathologies. These strategies enable more selective drug delivery, potentially enhancing therapeutic outcomes while mitigating peripheral toxicity. Clinical applications of these technologies are still predominantly in preclinical stages, with many studies conducted in animal models. Despite challenges in overcoming the BBB and ensuring the safety of nanocarriers, future developments are expected to expand the library of prodrugs and introduce novel approaches for binding BBB enhancers to other chemical entities, advancing the field of CNS drug delivery [109,116,117].

### 4.3. Neuroactive Peptides

Peptides are bioactive molecules with high specificity for potential targets. Protein-to-protein interaction can be stable and selective, which makes peptides a promising pharmacological target for therapy. However, several disadvantages associated with peptides have limited their clinical use. Firstly, peptides and proteins can act as antigens, triggering an immune response. Therefore, peptides tend to be immunogenic. Additionally, peptides are prone to degradation by various enzymes in the body, such as proteases and peptidases. This degradation poses two significant challenges: rapid compound breakdown and difficulty in oral administration.

Some alterations can be made in peptide structures to improve pharmacological and toxicological properties. First, amino acid residues such as Met, Ser, Ala, Thr, Val, and Gly at the N-terminus make the peptides more resistant to protease degradation [118]. Methionine-rich peptides also showed increased antimicrobial activity [119,120]. Also, C-terminal amidation and unnatural amino acid residues, e.g., D-enantiomers, improve stability [121]. D-peptides were developed against prion proteins, such as amyloid-β and Tau, associated with Alzheimer’s disease [122,123].

In order to increase affinity, stability, and binding affinity, a variety of cyclic peptides were developed [124]. One study showed improved pharmacological properties of CycloAnt, a cyclic peptide designed from a selected mu-opioid ligands library. CycloAnt demonstrated an antinociceptive effect without respiratory depletion and decreased hyperlocomotion in intraperitoneal administration [125]. Another study revealed that cyclotide, a natural cyclic peptide isolated from *Oldenlandia affinis*, showed immunosuppressant properties in an encephalomyelitis mouse model in oral administration [126].

The design and optimization of peptides to overcome degradation and low permeability have advanced significantly with the development of computational techniques [127]. In silico methods, such as molecular dynamics and docking (and others described above), have enhanced the selectivity of target–compound interactions. Regarding pharmacokinetics, the algorithms and software available for predicting the overall ADME properties of peptides remain far scarcer than those for small molecules. Although a few models have been successfully developed, these models were not designed to be general-purpose, meaning that each peptide type would typically require a separate study [121].

The in vitro and in vivo pharmacokinetics evaluation of peptides is still necessary and valuable. The data generated by these studies are evaluated using mathematical and statistical approaches existent in software like R, MATLAB^®^, and WinNonlin^®^ [128,129]. In vitro stability assays with human plasma, liver microsomes, and stomach fluid are commonly employed to assess the degradation process.

Moreover, in vitro techniques, such as mirror phage display, improved the experimental selection. These studies serve as the initial screening processes to evaluate the pharmacological efficacy of peptides. The phage display technique was developed to select a high-affinity peptide that binds to its targets. The idea is to use bacteriophages that produce the interested peptide via plasmid integration. This process comes from a bacteriophage library. These phages are then selected against the ligand and amplified [130,131]. Two peptides were selected by phage display for cancer therapy based on different targets. DPPA-1 (synthetic D-peptide that is a programmed death-1 (PD-1) and programmed death-ligand 1 (PD-L1) interaction antagonist/blocker with anticancer efficacy in vitro and in vivo) was selected against programmed death ligand 1, disrupting the binding with the receptor [132]. Similarly, the chosen d-PI_4 disrupted epidermal growth factor interaction with its receptor [133]. In addition, other libraries such as mRNA, ribosome, and cell surface display are used for peptide selection [134,135,136]. These methods follow similar steps but are tailored to specific needs and applications.

In-depth exploration of peptides as potential treatments for CNS disorders introduces an additional challenge: the BBB. Some peptide-development software is planned to predict this property, e.g., BBB-PEP https://github.com/Ansar390/BBB-PEP-Prediction/tree/main (accessed on 30 November 2024), and B3Pred https://webs.iiitd.edu.in/raghava/b3pred/ (accessed on 30 November 2024), and database collection, e.g., Brainpeps https://brainpeps.ugent.be/ (accessed on 30 November 2024) and B3Pdb https://webs.iiitd.edu.in/raghava/b3pdb/ (accessed on 30 November 2024) [137,138,139,140].

An approach for intracellular targets is alpha-helix structured peptides that can be inserted into the membrane [141]. Interestingly, most amphipathic or alpha-helix peptides are cytotoxic and are used for cancer and antimicrobial therapy. To address the BBB challenge, those cell-penetrating peptides have been identified as pharmacological therapy for CNS diseases. These peptides, such as transportan and TAT, can act as treatments and carriers of the active peptide portion through the BBB [142,143]. DK17, a modified peptide from penetratin, showed BBB permeation in silico and in vivo [144]. Cysteine-rich peptides, commonly found in spider venom, are also used as scaffolds for drug carriers [145,146]. Other techniques, like nanotechnology, have been employed, but they tend to make therapy more expensive and the bioavailability unpredictable.

## 5. Case Study

A relevant example of computational strategies applied to CNS drug design is the study by Srivastava et al. (2019) [147], which combined Gaussian-based QSAR modeling, MDo, MD simulations, and free energy calculations to optimize acetylcholinesterase (AChE) inhibitors with antioxidant properties. Their approach is an interesting reference for integrating in silico techniques to enhance BBB permeability predictions and improve drug design efficiency.

They used a Gaussian-based QSAR model to analyze structural features influencing AChE inhibition. The model was validated through cross-validation techniques, ensuring its reliability for guiding molecular modifications. The QSAR contour maps provided insights into steric, electrostatic, and hydrophobic influences, which helped refine molecular design choices.

Once key structural patterns were identified, MDo was employed to predict the binding interactions of the designed compounds with AChE. The docking results highlighted critical interactions at both the catalytic active site (CAS) and the peripheral anionic site (PAS), key regions involved in enzyme inhibition. Lead compounds, particularly compound 34, demonstrated strong π–π stacking and hydrophobic interactions, suggesting favorable binding conformations.

MM-GBSA free energy calculations ranked the compounds based on their estimated binding affinities to refine the predictions further. The most promising candidates were subjected to MD simulations to evaluate the stability of their interactions over a 50 ns period. Root mean square deviation and ligand–protein interaction analyses confirmed that compound 34 maintained a stable binding pose within AChE, reinforcing its potential as a CNS-active drug.

Additionally, this study incorporated in vitro and in vivo assays to validate computational findings. Notably, a parallel artificial membrane permeability assay (PAMPA-BBB) was used to assess BBB penetration, demonstrating that compound 34 exhibited permeability similar to donepezil. This experimental confirmation supported the in silico predictions and emphasized the practical application of computational models in CNS drug design.

This case study highlights how QSAR modeling, molecular docking, free energy calculations, and MD simulations can be effectively integrated to predict and optimize BBB-permeable compounds. The strategies employed align well with the objectives of this review, reinforcing the current importance of computational tools in guiding rational drug design for CNS applications.

## 6. Conclusions

The drug development process of small molecules or peptides requires the evaluation of the BBB permeation ability of the candidate. If the aim is to develop a compound capable of entering the brain or to ensure that it cannot diffuse through the BBB, the ability to predict this property is highly desirable, saving both time and resources. Several computational approaches are available to predict BBB crossing, including traditional QSAR/QSPR methods, which have been well-validated and refined, as well as more recent techniques that utilize artificial intelligence and machine learning. There is already a significant amount of scientific literature in this area, highlighting numerous encouraging studies. Even if these predictions are not flawless, the precision, accuracy, and dependence of existing methods are becoming increasingly sufficient for acceptance by regulatory agencies globally.

Integrative approaches will strengthen the BBB-permeable drug development. The use of predictive computational techniques with prodrug approaches within nanoparticle delivery systems has the potential to lead this research field.

Nonetheless, the complexity of BBB continues to be a challenge for predictive models. Factors like transporter-mediated absorption, efflux processes, and molecular-level physicochemical interactions are involved in the permeation process and are hard to simulate together, so there are no universal predictive models. Current machine learning techniques are promising since they can incorporate large and varied datasets. The effectiveness of this approach, however, depends on the data quality and good training sets that are still in development. Thus, there is an urgent need to increase experimental and computational datasets to improve these tools further.

## Figures and Tables

**Figure 1 pharmaceuticals-18-00217-f001:**
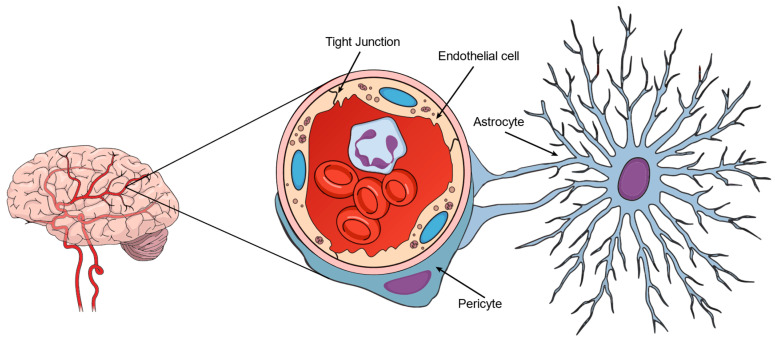
Schematic illustration of the blood–brain barrier morphology.

**Figure 2 pharmaceuticals-18-00217-f002:**
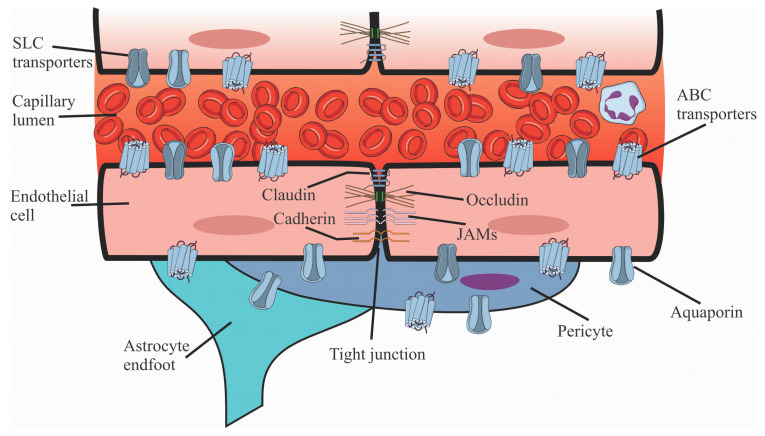
Schematic BBB. Brief representation of protein interaction associated with tight junctions (claudin, occludin, cadherin, and JAMs). Transporters associated with brain homeostasis are represented by solute carrier (SLC) transporters, ATP-binding cassette (ABC) transporters, and aquaporin.

**Figure 3 pharmaceuticals-18-00217-f003:**
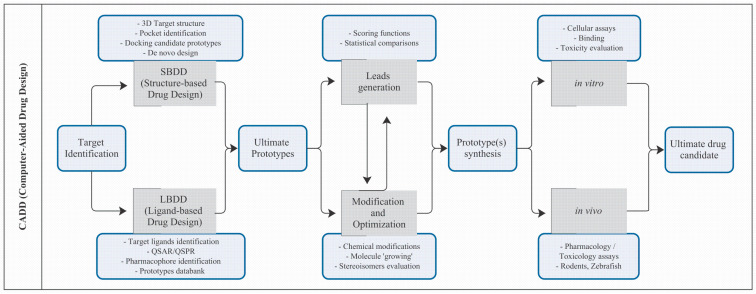
Drug candidate development using CADD planning based on SBDD and LBDD.

**Figure 4 pharmaceuticals-18-00217-f004:**
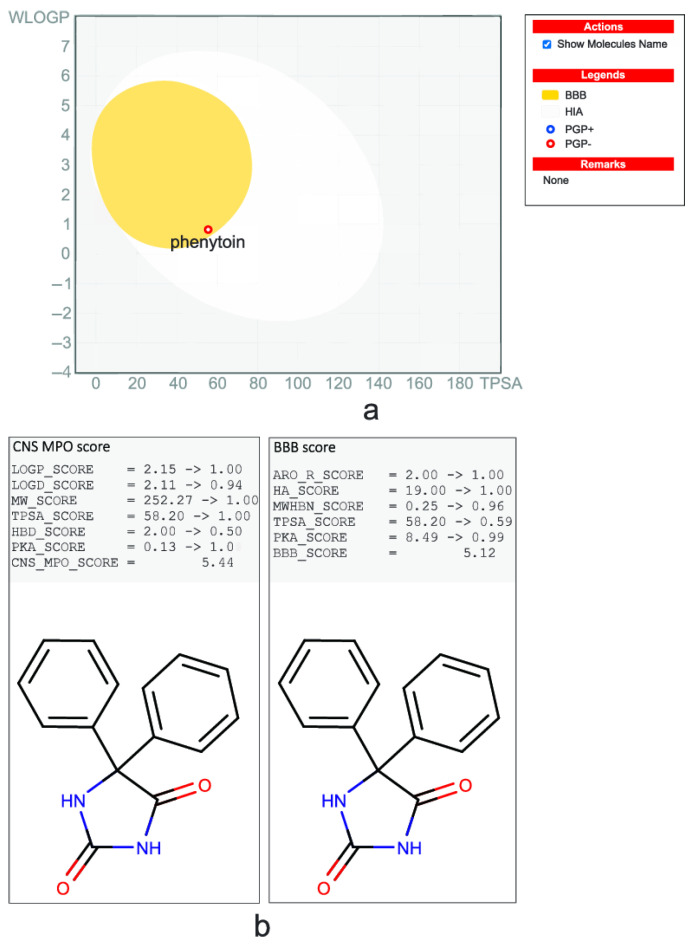
Evaluation of phenytoin (an antiepileptic drug known for its CNS action) BBB permeation. (**a**) “Boiled egg” permeation model (SwissADME); (**b**) Scores used by Chemaxon’s Marvin to predict the CNS permeation. HIA: human intestinal absorption; PGP: P-glycoprotein substrate.

**Figure 5 pharmaceuticals-18-00217-f005:**
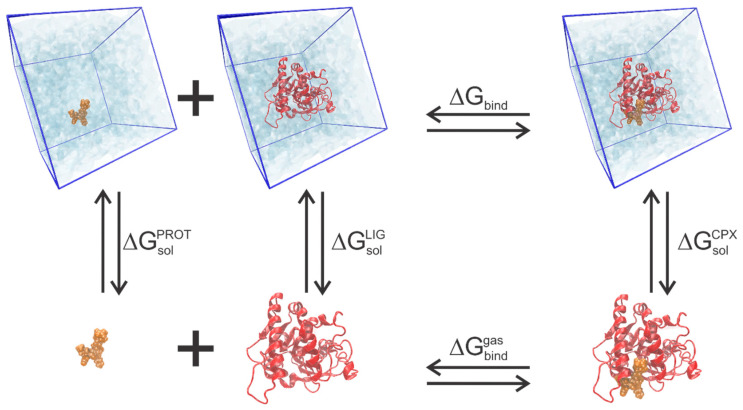
Thermodynamic cycle for binding free energy calculations.

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
