# Peer review of "Computational Modeling of Pharmaceuticals with an Emphasis on Crossing the Blood–Brain Barrier"

_pharmaceuticals, 2025, doi:10.3390/ph18020217_

Round 1

Reviewer 1 Report

Comments and Suggestions for Authors

Title: Computational modeling of pharmaceuticals with an emphasis on crossing the blood-brain barrier

The manuscript presents a comprehensive review of computational modeling techniques used for predicting pharmaceuticals’ ability to cross the BBB.

Below are recommendations to enhance the clarity and scientific value of the manuscript:

  1. While the manuscript summarizes current knowledge and methods to predict lipophilicity and other related properties of candidate compounds, it would benefit from including strategies to enhance BBB permeability. Authors should discuss advanced approaches such as prodrug design, nanoparticle delivery systems, or chemical modifications aimed at improving BBB penetration without compromising compound specificity or safety.
  2. The abstract currently provides general information about the topic. It should be revised to include specific highlights and key findings from the manuscript. Emphasizing innovative methodologies or significant conclusions will make the abstract more informative.
  3. The conclusion effectively summarizes existing knowledge and evaluation techniques for BBB crossing. To strengthen its impact, the authors should propose integrative insights or future directions based on the reviewed data.

By addressing these recommendations, the manuscript will significantly contribute to the field of CNS drug discovery and provide a valuable resource for researchers.

Author Response

Reviewer 1:

While the manuscript summarizes current knowledge and methods to predict lipophilicity and other related properties of candidate compounds, it would benefit from including strategies to enhance BBB permeability. Authors should discuss advanced approaches such as prodrug design, nanoparticle delivery systems, or chemical modifications aimed at improving BBB penetration without compromising compound specificity or safety.

  • Thank you for pointing this out. Even if the main focus of the manuscript is to summarize the computational methods involved in developing CNS drugs, the authors understand that these topics can significantly improve the scope of the publication. Therefore, we have included one new topic (4.1 and 4.2), lines 706 – 776 pages 15 and 16, highlighted in yellow.

The abstract currently provides general information about the topic. It should be revised to include specific highlights and key findings from the manuscript. Emphasizing innovative methodologies or significant conclusions will make the abstract more informative.

               We agreed and therefore changed the abstract to the following text:

“The discovery and development of new pharmaceutical drugs is a costly, time-consuming and highly manual process, with significant challenges in ensuring drug bioavailability at target sites. Computational techniques are highly employed in drug design, particularly to predict the pharmacokinetic properties of molecules. One major kinetic challenge in central nervous system drug development is the permeation through the Blood-Brain Barrier (BBB). Several different computational techniques are used to evaluate both BBB permeability and target delivery. Methods such as Quantitative Structure-Activity Relationships, Machine Learning models, Molecular Dynamics simulations, End-Point Free Energy calculations, or Transporter Models have pros and cons for drug development, and all contribute to a better understanding of a specific characteristic. Additionally, the design (assisted or not by computers) of prodrug and nanoparticle-based drug delivery systems can enhance BBB permeability by leveraging enzymatic activation and transporter-mediated uptake. Neuroactive peptide computational development is also a relevant field in drug design, since biopharmaceuticals are on the edge of drug discovery. By integrating these computational and formulation-based strategies, researchers can enhance the rational design of BBB-permeable drugs while minimizing off-target effects. This review is a valuable resource for understanding BBB selectivity principles and the latest in silico and nanotechnological approaches for improving CNS drug delivery.” Lines 12 – 27, Page 1, highlighted in yellow.

The conclusion effectively summarizes existing knowledge and evaluation techniques for BBB crossing. To strengthen its impact, the authors should propose integrative insights or future directions based on the reviewed data.

We agreed and therefore added to the conclusion the following statement:

“Integrative approaches will strengthen the BBB-permeable drug development. The use of predictive computational techniques with prodrug approaches within nanoparticle delivery systems has the potential to lead this research field.” Lines 895-897, Page 19, highlighted in yellow.

Reviewer 2 Report

Comments and Suggestions for Authors

In the manuscript entitled "Computational Modeling of Pharmaceuticals with an Emphasis on Crossing the Blood-Brain Barrier," the authors provide a brief review of computational methods to discover new drugs that can cross or not the BBB. It's an exciting work due to the importance of the use of computational methods in drug design and the need to identify methods of prediction of new drugs against the central nervous center or to avoid side effects. However, for the acceptance, the authors must pay attention to the following aspects:

- In the abstract, the authors provide the following information: "This review will comprehensively address the BBB." The sentence needs complementation. What's aspects explored about BBB? The authors are encouraged to improve.

-  The abstract must be improved. It looks like a brief introduction, not a summary of the work. Thus, the authors are encouraged to improve and add information about the conclusions about computational methods used to identify aspects that can be used in designing new drugs targeting BBB or not that cross the BBB.

- The introduction needs a paragraph about computational methods in drug design and its use to discover drugs that cross or not the BBB. Please read and cite the following reference: https://www.eurekaselect.com/article/121637

- The final part of the introduction needs the work objectives. The authors are encouraged to add this information.

- In the topic "3.1. Computer-Aided Drug Design Methods," authors are encouraged to remove the dashes and modify them to text format.

- The manuscript needs a topic about the challenges and opportunities of CADD and BBB to improve the quality of the work.

- It would be interesting if the authors added a study of the case, describing some drug that can cross the BBB discovered by computational methods of molecular docking and molecular dynamics simulations or using the endpoints methods. I am Missing a more in-depth description of drugs or even prototype compounds.

Author Response

Reviewer 2

In the manuscript entitled "Computational Modeling of Pharmaceuticals with an Emphasis on Crossing the Blood-Brain Barrier," the authors provide a brief review of computational methods to discover new drugs that can cross or not the BBB. It's an exciting work due to the importance of the use of computational methods in drug design and the need to identify methods of prediction of new drugs against the central nervous center or to avoid side effects. However, for the acceptance, the authors must pay attention to the following aspects:

- In the abstract, the authors provide the following information: "This review will comprehensively address the BBB." The sentence needs complementation. What's aspects explored about BBB? The authors are encouraged to improve.

-  The abstract must be improved. It looks like a brief introduction, not a summary of the work. Thus, the authors are encouraged to improve and add information about the conclusions about computational methods used to identify aspects that can be used in designing new drugs targeting BBB or not that cross the BBB.

Thank you for your comments. Indeed, following the recommendation of the other reviewer, we provided a new abstract; please evaluate if it now has the required structure and information.

- The introduction needs a paragraph about computational methods in drug design and its use to discover drugs that cross or not the BBB. Please read and cite the following reference: https://www.eurekaselect.com/article/121637

Unfortunately, we don’t have access to this reference. We asked the editorial personnel, but they were not able to provide the full text as well. Anyway, we agreed with the recommendation and included one paragraph at the end of the introduction (lines 63 – 66, page 2, highlighted in yellow ).

- The final part of the introduction needs the work objectives. The authors are encouraged to add this information.

We agreed, therefore we added the information “This review aims to provide a comprehensive overview of the BBB and the computational techniques involved in developing CNS-targeted drugs”. (lines 66 – 68, page 2, highlighted in yellow)

- In the topic "3.1. Computer-Aided Drug Design Methods," authors are encouraged to remove the dashes and modify them to text format.

Done, the topic is now in text format (lines 179 – 228, pages 5 and 6, highlighted in yellow).

- The manuscript needs a topic about the challenges and opportunities of CADD and BBB to improve the quality of the work.

Thank you for pointing this out. Indeed, the topic is relevant. We added item 3.1.5 (lines 659 – 704, pages 14 and 15, highlighted in yellow)

- It would be interesting if the authors added a study of the case, describing some drug that can cross the BBB discovered by computational methods of molecular docking and molecular dynamics simulations or using the endpoints methods. I am Missing a more in-depth description of drugs or even prototype compounds.

Agreed. We included a Case Study of the paper published in 20019 by Dr Srivastava, focusing on the methods they used and on their results. (lines 847 – 882, page 18)

Round 2

Reviewer 1 Report

Comments and Suggestions for Authors

Authors have adequately addressed the issues raised in the review. Therefore, I recommend accepting the manuscript for the publication in Pharmaceuticals.

Reviewer 2 Report

Comments and Suggestions for Authors

Now, the manuscript can be accepted. Congratulations to the authors.